# PIK3CAMutations in Breast Cancer Subtypes Other Than HR-Positive/HER2-Negative

**DOI:** 10.3390/jpm12111793

**Published:** 2022-10-31

**Authors:** Liliana Ascione, Paola Zagami, Eleonora Nicolò, Edoardo Crimini, Giuseppe Curigliano, Carmen Criscitiello

**Affiliations:** 1Division of New Drugs and Early Drug Development for Innovative Therapies, European Institute of Oncology, IRCCS, Via Ripamonti 435, 20141 Milan, Italy; 2Department of Oncology and Haematology (DIPO), University of Milan, Via Festa del Perdono 7, 20122 Milan, Italy; 3Lineberger Comprehensive Cancer Center, University of North Carolina at Chapel Hill, Chapel Hill, NC 27599, USA

**Keywords:** *PIK3CA*, triple negative breast cancer, HER2 positive breast cancer, targeted therapy, PI3K inhibitors

## Abstract

The phosphoinositide 3-kinase (PI3K) pathway plays a key role in cancer, influencing growth, proliferation, and survival of tumor cells. *PIK3CA* mutations are generally oncogenic and responsible for uncontrolled cellular growth. PI3K inhibitors (PI3Ki) can inhibit the PI3K/AKT/mTOR pathway, although burdened by not easily manageable toxicity. Among PI3Ki, alpelisib, a selective p110α inhibitor, is approved for the treatment of hormone receptor (HR)+/HER2- *PIK3CA* mutant metastatic breast cancer (BC) that has progressed to a first line endocrine therapy. *PIK3CA* mutations are also present in triple negative BC (TNBC) and HER2+ BC, although the role of PI3K inhibition is not well established in these subtypes. In this review, we go through the PI3K/AKT/mTOR pathway, describing most common mutations found in PI3K genes and how they can be detected. We describe the available biological and clinical evidence of *PIK3CA* mutations in breast cancers other than HR+/HER2-, summarizing clinical trials investigating PI3Ki in these subtypes.

## 1. Introduction

The phosphoinositide 3-kinase (PI3K) pathway plays a key role in cancer, influencing growth, proliferation, and survival of tumor cells [1]. Activation of this pathway is observed in approximately 70% of all breast cancer (BC) cases. The *PIK3CA* gene encodes for the catalytic alpha subunit (p110α) of class I PI3K and is commonly mutated in human cancers, constantly stimulating tumor growth and survival [2]. *PIK3CA* mutations occur in 20–30% of patients with BC, with different prevalence among the various BC intrinsic subtypes, reported in approximately 45% of luminal A, 29% of luminal B, 39% of HER2-enriched and in 9% of basal-like BC [3]. *PIK3CA* mutations have been associated with a better disease-free survival (DFS) in early hormone receptor positive (HR+)/HER2-negative (HER2-) BC [4], while *PIK3CA* mutant HR+/HER2- metastatic BC (mBC) showed resistance to chemotherapy and poorer outcomes [5]. PI3K is a druggable target, with pan-PI3K inhibitors (i.e., active against all the PI3K isoforms, such as buparlisib) as well as isoform-specific PI3K inhibitors (such as alpelisib, a p110α specific inhibitor) being available [1].

The existing crosstalk between the estrogen receptor (ER) and the PI3K-pathways represent the biological rationale for investigating the implications of *PIK3CA* gene alterations in HR+/HER2- BC. The uncontrolled growth of HR+/HER2- BC is partly dependent on hormone receptors (HR) signaling pathways, involved in the transcription of growth-related genes (such as insulin growth factor I receptor, *IGF-1R*) and successfully inhibited by endocrine therapies (ET). The observed association between PI3K pathway alterations, including *PIK3CA* mutations, and the development of endocrine resistance as well as the enhancement of ER pathway induced by PIK3CA inhibitors in *PIK3CA*-mutated tumors are the basis for combining anti-endocrine therapy and PIK3CA inhibitors [6,7].

Specifically, the role of targeting *PIK3CA* mutations in HR+/HER2- mBC has been established according to the results of the phase III trial SOLAR-1 which demonstrated a statistically significant improvement in progression-free survival (PFS) by the addition of alpelisib, a PI3Kα-specific inhibitor, to fulvestrant in post-menopausal patients with *PIK3CA* mutated, HR+/HER2- mBC who progressed to prior ET [8]. Accordingly, most recent ESMO guidelines for mBC management recommend testing for *PIK3CA* mutations, to identify the best second-line treatment in patients with HR+/HER2- mBC progressing to first-line standard of care (SoC) [9].

Going beyond HR+/HER2- BC, here we aim to analyze the biology and clinical implication of *PIK3CA* mutations in HER2-positive (HER2+) and triple-negative BC (TNBC), focusing on preclinical and clinical evidence. Furthermore, we discuss available data from trials investigating PI3K inhibitors in these BC subtypes, providing an overview of the ongoing clinical trials.

## 2. The PIK3CA/AKT/mTOR Pathway

There are three different classes of phosphoinositide 3-kinases (PI3Ks): class I, II and III. Class I PI3Ks are heterodimers made up of a catalytic subunit, which activity is modulated by a regulatory one.

The catalytic subunit (p110) is expressed in four different isoforms, according to which class I PI3Ks are divided into two sub-groups: class IA (p110α, p110β, p110δ—encoded by *PIK3CA*, *PIK3CB* and *PIK3CD* genes, respectively) and class IB (p110γ, encoded by *PIK3CG* gene) [10]. Structurally, class IA and class IB catalytic subunits have five and four different domains, respectively.

Class IA regulatory subunits, generally called “p85”, are encoded by three genes, *PIK3R1* (p85α; p55α; p50α), *PIK3R2* (p85β) and *PIK3R3* (p55γ). p85 has five different domains, of which Src homology 2 (SH2) ones mediate the interaction with the p110 catalytic subunit, binding its C2, the helical and the kinase domains maintaining the enzyme in a low activity status [11].

PI3K is activated by the binding between growth factors and their tyrosine kinase receptors (RTKs) or, indirectly, via adaptor molecules (e.g., insulin receptor substrate 1, IRS1). Once stimulated, RTKs and adaptor proteins expose phosphorylated tyrosine residues, bound by the SH2 domain of the regulatory subunit, that relieves the inhibition of the catalytic subunit, facilitating its stimulation by other molecules, especially RAS (rat sarcoma virus oncoprotein homolog).

Once activated and close to the cell membrane, the catalytic subunit phosphorylates the cell membrane lipid phosphatidyl inositol 4,5 bisphosphate (PIP2) to phosphatidyl inositol 3,4,5 triphosphate (PIP3), by adding a phosphate on the 3’OH position. PIP3 acts as a second messenger and recruits downstream elements containing a pleckstrin homology (PH) domain, such as AKT (protein kinase B, PKB), a serine-threonine kinase, and PDK1 (3-phosphoinositide-dependent protein kinase-1) at the cell membrane level. Once close to each other, PDK1 activates AKT, phosphorylating it at threonine 308 [12,13].

Activated-AKT can inhibit the forkhead (FOXO) family of transcription factors by phosphorylation, promoting the G1-S cell-cycle transition and blocking the FOXO-mediated transcription of pro-apoptotic proteins such as Fas-ligand (FasL) and Bim (BCL2-like 11), as well as hindering p53-mediated apoptosis. In addition, AKT leads to the activation of the mammalian target of rapamycin complex 1 (*mTORC1*), through the phosphorylation of the tuberous sclerosis complex 2 (TSC2)*,* thus enhancing protein synthesis by increased activity of the eukaryotic translation initiation factor 4E and of the p70 S6 kinase (p70S6K).

At the same time, the mammalian target of rapamycin complex 2 (*mTORC2*) provides the full activation of AKT through the S473 residue phosphorylation. Of note, p70S6K contributes at reducing the PI3K/AKT pathway signaling by inhibiting the insulin receptor substrate 1 (IRS1), an adaptor protein normally activated by the insulin growth factor 1 (IGF1).

PI3K signaling is principally antagonized by phosphatase and tensin homolog (PTEN), a lipid phosphatase responsible for the dephosphorylation of PIP3 into PIP2, thereby terminating signal transmission [10,14] [Figure 1].

## 3. *PIK3CA* Mutations and Their Detection

Somatic point mutations and gene amplifications are the two main genetic alterations that can alter *PIK3CA* functions [15,16].

Wu et al. [16] found a *PIK3CA* gene copy number gain in 8.7% BC over 92 cases evaluated in their study, an interesting finding since *PIK3CA* gene amplification has been described as a mechanism of resistance to selective PIK3CA inhibitor in HER2+, *PIK3CA* mutant breast cancer cell line, KPL-4 [17].

About 80% of *PIK3CA* somatic mutations are hot-spot mutations, the most common being single nucleotide changes resulting in aminoacidic substitutions: E542K and E545K in the exon 9, H1047R and H1047L in the exon 20. These mutations are oncogenic, determining a gain of function that promotes the constitutive signaling pathway activation, no longer dependent on growth factor stimulation. Of note, *PIK3CA* mutations have been found in ductal carcinoma in situ (DCIS), suggesting that it is an early event in the BC tumorigenesis [18].

Exon 9 mutations involve the p110 helical domain, avoiding its inhibitory interaction with the p85 nSH2 domain, mimicking the activation by RTK phosphopeptides. These mutants depend on Ras binding for their transformation [19].

By contrast, exon 20 mutations at the hotspot 1047 do not need Ras activation but depend on p85 binding. These mutations occur in the kinase domain causing a conformational change of the activation loop, thus favoring the lipid membrane binding and the PIP2 substrate recognition [20] [Figure 1].

The most three frequent mutations E542K, E545K and H1047R are all responsible for enzymatic activation of the PI3Kα; however, in mice models, *PIK3CA* H1047R has shown to be a stronger inducer of breast cancer, in terms of both latency and frequency, maybe explained by the higher activation of the downstream AKT signaling [21].

PI3K pathway is also involved in the cell-signaling of the family of Human epidermal growth factor receptors (HER/erbB), including the HER2 that is overexpressed in HER2+ BC subtype [22,23]. In cell membrane, the monomer HER2 is inactive. Following the binding of the ligand, HER2 hetero- or homo-dimerizes to become active, stimulating the downstream PI3K/AKT and the mitogen-activated protein kinase (MAPK) pathway, inducing cell growth and survival.

The aforementioned mutations in *PIK3CA* gene or partial/complete loss of PTEN, mediating mTOR activation, may lead to HER2-independent constitutive activation of the PI3K/AKT pathway that has been associated with HER2-therapy resistance [24,25]. In HR+/HER2+ BC, PI3K is involved in both ER and HER2 pathway signaling, and crosstalk among these pathways is responsible for the occurrence of endocrine- and HER2-resistance [26].

There are three main molecular biology techniques available for the detection of *PIK3CA* mutations: Sanger sequencing, real-time PCR (RT-PCR) and next-generation sequencing (NGS).

Sanger sequencing represents a reliable method that can detect unknown mutations and gene fusions, although limited by a lower sensitivity when compared to NGS. RT-PCR is a commonly employed technique to detect mutations in known genomic regions; however, using primers for specific mutant sequences, only a restricted number of already well-characterized alterations can be assessed. On the other hand, NGS is a high throughput technique for the analysis of tumor DNA samples with higher sensibility than Sanger sequencing, so possibly representing the “ideal technique”, but its adoption is limited because of the complexity of the procedure and, importantly, its costs [27,28].

The method used in the SOLAR-1 trial is the Therascreen © PIK3CA RGQ PCR kit, a real time qualitative PCR test that can identify 11 mutations, in particular exon 7 p.C420R; exon 9 p.E542K, p.E545A, p.E545D, p.E545G, p.E545K, p.Q546E and p.Q546R; exon 20 p.H1047L, p.H1047R and p.H1047Y. It analyses both genomic DNA extracted from formalin-fixed, paraffin-embedded tumor tissue or circulating tumor DNA (ctDNA) from plasma derived from liquid biopsies. Along with the approval of alpelisib for the treatment of *PIK3CA* mutated-mBC, the FDA approved the Therascreen © PIK3CA RGQ PCR kit as the companion diagnostic (CDx) for the detection of *PIK3CA* mutations [29].

Martines-Saez et al. selected a dataset of 6338 BC tumor samples analyzed by targeted or whole exome sequencing. Exon 20 p.H1047R and exon 9 p.E545K and p.E542K represented approximately 63% of all the *PIK3CA* mutations found. Nevertheless, other *PIK3CA* mutations have been found in a non- negligible frequency, such as the exon 4 p.N345K. Since these mutations are not detected in the Therascreen © kit, the SOLAR1 trial did not investigate the efficacy of alpelisib against these variants and this remains unknown [30].

This issue should be considered when choosing a platform for mutational status detection, especially because wide-spectrum NGS assays can detect more alterations beyond those included in the Therascreen. In fact, in the U.S., along with the Therascreen © kit, FoundationOne^®^ CDx and FoundationOne^®^ Liquid CDx are approved as companion diagnostics too.

Regarding samples, *PIK3CA* mutational status can vary among primary tumor and metastases, as for the HR and the HER2 expression [31]. The discordance has been highlighted in different studies, not only in terms of gain or loss of the mutation in the *PIK3CA* gene but also in terms of increasing or decreasing levels of mutation [32,33,34,35], possibly influencing response to PI3Ki and underlining the importance of molecular characterization of metastatic sites. Moreover, if specimens cannot be obtained (for example due to the specific metastatic site), ctDNA can be used, selecting patients whose tumor shedding is expected to be sufficient to avoid false negative (e.g., cases with adequate tumor burden or with progressive disease) [36]. Among 68 patients included in a study by Dumbrava et al. [37], an agreement rate of 78% in the detection of *PIK3CA* mutations has been demonstrated between ctDNA samples and tumor tissue analysis. The concordance even increases up to 91% if only patients with progressing tumors are considered [37].

## 4. Targeting PI3K

PI3Ki include pan-PI3K inhibitors (e.g., buparlisib, pilaralisib, pictilisib), isoform-specific inhibitors (e.g., alpelisib, inavolisib, idelalisib) or dual PI3K/mTOR inhibitors (e.g., dactolisib). They are mainly reversible and ATP-competitive inhibitors since they compete with ATP for the anchoring to the “ATP binding site” of the catalytic subunit.

The anti-tumor effects related to the blocking of PI3K signaling is not only due to the inhibition of the pro-survival mechanism mediated by its downstream effectors (e.g., AKT or mTORC1) but also associated with tumor microenvironment modulation. In fact, PI3K favors tumor angiogenesis and evidence shows that p110α inhibition can affect formation of neo-vessels. PI3K isoforms are expressed widely in the human organism: PI3Kα and PI3Kβ are the most ubiquitously expressed isoforms, while PI3Kγ and PI3Kδ are mostly located in leukocytes, having a role in the immune-response control. Specifically, PI3Kγ contributes at suppressing T-cells in mouse models, while PI3Kδ regulates the antigen receptor signaling in lymphocytes; their inhibition plays a role in reprogramming the immune system against solid tumors [38].

The only well recognized biomarker predictive of response to alpelisib is the presence of *PIK3CA* mutations and seems to be independent of the presence of the “PI3K pathway activation”, meaning that alpelisib provides a clinical benefit even in tumors harboring *PIK3CA* mutations that do not cause a downstream activation of the PI3K/AKT/mTOR pathway. On the contrary, PTEN loss could be related to alpelisib resistance [39,40]. Of note, tumors with double *cisPIK3CA* mutations proved to be more sensitive to PI3Kα-inhibitors than single hotspot mutations, supporting the hypothesis of a synergistic effect of the multiple hotspot hits. Conversely, the presence of *PIK3CA* copy number gains in addition to a *PIK3CA* mutation have not been associated with better responses to alpelisib in a small group of BC patients [41,42].

Regarding tolerability, pan-PI3K inhibitors (pan-PI3Ki), targeting all the PI3K class I isoforms (PI3Kα, PI3Kβ, PI3Kγ and PI3Kδ), are associated with higher risk of on-target and off-target toxicities. The toxicity profile of pan-PI3Ki has severely limited their clinical use, due to the adverse events (AEs) described in different trials with buparlisib that led to discontinuation of its clinical development [43]. Most reported pan-PI3Ki-related side effects are hyperglycemia, rash, neutropenia, hepatotoxicity, diarrhea and neuropsychiatric effects, such as depression or anxiety—confirming the capacity of these drugs to cross the haemato-encephalic barrier.

Isoform-specific antagonists have been developed to improve the safety profile of PI3K-targeting drugs, enabling tumor growth inhibition while maximizing tolerability.

AEs observed with these agents reflect the distribution of the target isoform and its role in the organism. According to data reported from the SOLAR-1 trial, the most frequent AE related to alpelisib is hyperglycemia (65%), explained by the involvement of the PI3K/AKT pathway in glucose metabolism: AKT favors the expression of glucose transporters (GLUT4) in muscle and fat cells, while blocking liver gluconeogenesis. When inhibited, blood glucose levels increase, causing hyperinsulinemia that stimulates IRS1, contributing to activating the PI3K/AKT pathway and possibly representing a mechanism of resistance to anti-PI3K drugs. Hyperglycemia generally occurs within the first two cycles, and is managed with metformin or, if not sufficient, insulin-sensitizer and/or insulin therapy. It has been the most common grade (G) 3/G4 event that led to dose reduction or discontinuation of alpelisib. Another frequent AE related to alpelisib, usually milder, is diarrhea (58%), treated with changes in eating habits and, if persistent, with loperamide (1st line) or octreotide (2nd line). Since the PI3K/AKT pathway is involved in the keratinocyte differentiation, rash is a common AE reported in the SOLAR-1 (54%), reversible when treated early with topical steroids and antihistamines in the case of pruritus. Although typically associated with mTOR inhibitors, stomatitis has been registered in patients treated with alpelisib (25%) and can be prevented with topical steroids. If AEs are not manageable, discontinuation of alpelisib is recommended, as in the case of pneumonitis, a rare AE reported in 2% of cases in the SOLAR-1 trial [8,44].

Considering other isoforms, PI3Kγ is expressed in vascular smooth muscle cells and seems to play a role in the myogenic tone regulation, possibly causing hypertension when inhibited. As PI3Kγ is also involved in the immune system regulation along with PI3Kδ, their hindering can elicit auto-immune phenomena, such as auto-immune colitis [45].

Although targeting single isoforms can abate the toxicity spectrum, optimizing isoform specific-PI3Ki faces some difficulties, especially related to the high homology of the ATP-binding sites of all the class I PI3Ks. Of note, they only differ for a few “non conserved residues”, mainly in the region near the hinge and in the phosphate binding loop (P-loop) of the kinase domain [46].

## 5. Biological and Clinical Evidence of *PIK3CA* Mutations in HER2-Positive Breast Cancer

HER2 overexpression/amplification occurs in approximately 15–20% of all BCs, representing one of the most aggressive BC subtypes, characterized by rapid tumor growth, visceral and brain metastases onset.

Therapeutic regimens approved for HER2+ BC include anti-HER2 monoclonal antibodies (mAbs) (such as trastuzumab, pertuzumab and margetuximab), or tyrosine kinase inhibitors (TKIs) (such as lapatinib, neratinib and tucatinib), generally combined with chemotherapy, and antibody drug conjugates (ADCs) such as trastuzumab emtansine (T-DM1) and trastuzumab deruxtecan (T-DXd) [47].

About 30–40% of HER2+ BC cases carry a *PIK3CA* mutation, and such mutations have been associated with trastuzumab resistance [48]. Two phase III trials, BOLERO-1 and BOLERO-3, were conducted to evaluate the efficacy of the mTOR inhibitor everolimus in combination with trastuzumab and chemotherapy in patients with HER2+ mBC. In the joint analysis of these two studies, the addition of everolimus provided a statistically significant PFS benefit in patients with *PIK3CA* tumor mutations in the ER- subgroup (hazard ratio = 0.43; 95% CI, 0.22–0.86), with a lower effect in the ER+ subgroup (hazard ratio = 0.93; 95% CI, 0.55–1.57). These results are consistent with a crosstalk between the ER, HER2 and PI3K signaling pathways [49].

Several biomarker analyses evaluated the role of *PIK3CA* mutations in influencing treatment response in early and metastatic setting. In 2016, Loibl et al. published a pooled analysis of data deriving from different clinical trials (GeparQuattro [50], GeparQuinto [51], GeparSixto [52], NeoALTTO [53] and CHERLOB [54]) to assess whether *PIK3CA* mutations could impact the achievement of pathologic complete response (pCR) in HER2+ BC treated with neoadjuvant anti-HER2 agents in association with chemotherapy. Secondary endpoints of this analysis include the association with disease-free survival (DFS) and overall survival (OS). Overall, pCR was significantly lower in the *PIK3CA* mutant cohort when compared to the wild-type one, especially in HR+ tumors; however, no significant differences in terms of DFS or OS were observed [55]. Considering the adjuvant setting, biomarker analysis of the FinHER [56] and of the NSABP-B31 [57] did not show significant interaction between *PIK3CA* status and trastuzumab benefit, nor differences in DFS among *PIK3CA* wild-type or mutant BC [58]. On the contrary, Cizkova et al. [59] found poorer DFS and OS in *PIK3CA*-mutant HER2+ BC treated with neoadjuvant and/or adjuvant trastuzumab, compared with wild type HER2+ BCs [59]; similarly, a detrimental impact of *PIK3CA* mutation on adjuvant treatment outcome in HER2+ BC has been reported by Jensen et al. [60].

Regarding mBC, some data concerning the role of *PIK3CA* mutations in HER2+ BC derive from biomarker analyses of the CLEOPATRA trial and the EMILIA trial. As for the CLEOPATRA trial, a study that evaluated the addition of pertuzumab to trastuzumab and docetaxel in first-line treatment of HER2+ mBC, a significantly worse prognosis was found in patients with *PIK3CA*-mutant tumors regardless of the treatment arm [61]. According to Baselga et al., in the EMILIA trial, which compared capecitabine plus lapatinib to T-DM1 in second line, *PIK3CA* mutant HER2+ tumors had a lower sensibility to lapatinib and chemotherapy, while T-DM1 seemed to be effective in both *PIK3CA* mutant and wild type tumors [62]. This can be explained by the peculiar mechanism of action of ADCs: while mAbs and TKIs disrupt intracellular HER2 signaling, ADCs exploit HER2 to allow a targeted release of highly cytotoxic drugs, regardless of downstream signals [63].

All these findings form a basis for designing clinical trials investigating anti-HER2 agents in combination with PI3K inhibitors (PI3Ki), to elucidate whether blocking *PIK3CA* signaling can somehow improve clinical outcomes in HER2+/*PIK3CA* mutant BC, while better defining the role of *PIK3CA* mutations in HER2+ BC.

Buparlisib has been evaluated in combination with trastuzumab or lapatinib in patients with trastuzumab-resistant mBC, unselected for *PIK3CA* mutation. As for the first combination, among 17 patients treated with buparlisib and trastuzumab, treatment was well tolerated, with rash (39%), hyperglycemia (33%), and diarrhea (28%) being the most common AEs. Regarding psychiatric disorders, such as anxiety and depression, two patients developed G3 drug-related psychiatric AEs, not leading to treatment discontinuation. The observed disease control rate (DCR) was 59% and the clinical benefit rate (CBR) was 18% [64], suggesting that PI3K inhibition with buparlisib can restore sensitivity to trastuzumab. Nevertheless, since buparlisib has also shown activity in a small group of patients with mBC as a monotherapy [65], it is hard to define to which extent the observed activity in this trial is related to buparlisib itself. Despite these initial encouraging results, a phase II study of buparlisib and trastuzumab in HER2+ mBC demonstrated limited efficacy, regardless of *PIK3CA* mutations, with an overall response rate (ORR) of 10%, thereby failing to meet the primary endpoint (ORR ≥ 25%). The safety profile was consistent with that reported in the phase Ib part of the trial [66].

The phase Ib/II PIKHER2 study evaluated the combination of buparlisib with lapatinib in HER2+ trastuzumab-resistant mBC. Main TRAEs (treatment-related adverse events) leading to discontinuation of treatment were skin rash, gastrointestinal disorders, depression and anxiety. The DCR was 79%, including one patient experiencing a complete response, and the CBR was 29% [67].

Although the clinical benefit reported in the aforementioned trials was slightly numerically higher in patients with *PIK3CA*-mutated tumors, responses were observed also in patients with *PIK3CA* wild-type tumors. Of note, preliminary signs of activity have been reported in patients with brain metastases treated with buparlisib, a common site of metastases in HER2+ BC.

In the NeoPHOEBE phase II trial, buparlisib was evaluated in the neoadjuvant setting: patients with HER2+ early BC, regardless of *PIK3CA* mutation, were randomized to receive trastuzumab plus either buparlisib or placebo for six weeks, followed by trastuzumab and paclitaxel plus buparlisib or placebo. There was no significant difference in the pCR rate (32% vs. 40%; *p* = 0.811) between the buparlisib and the placebo arm; however, a trend towards higher ORR at six weeks (68.8% vs. 33.3%; *p* = 0.053) was observed in the ER+ subgroups, supported by a significant decrease in Ki-67 (75% vs. 26.7%; *p* = 0.021). Only eight patients were *PIK3CA* mutant, limiting the ability of this study to detect differences in the *PIK3CA* wild type/mutant cohorts. Enrollment in this study was suspended earlier, after the inclusion of 50 of the 256 planned patients, mainly because of liver toxicity (G ≥ 3 ALT and AST increase in 48% and 28% with buparlisib compared to 8% and 0% with placebo, respectively) [68].

Pilaralisib, another pan-PI3Ki, has been evaluated, in combination with trastuzumab or trastuzumab plus paclitaxel, in a phase I/II study enrolling 42 trastuzumab-resistant HER2+ mBC. The most frequent TRAEs were diarrhea, fatigue and rash with a higher frequency of AEs in the paclitaxel arm. Responses were observed only in the chemotherapy one (20%, all partial responses) and did not correlate with *PIK3CA* mutations in cell-free circulating DNA [69]. Considering that all patients in the paclitaxel arm had received prior taxanes and trastuzumab, a benefit from the addition of pilaralisib is suggested, supporting the preclinical evidence of a synergy between chemotherapy and PI3K pathway inhibition [70,71].

Copanlisib, an intravenously administered pan-PI3Ki, mainly active against PI3Kα and PI3Kδ isoforms, has been studied in combination with trastuzumab, in the phase Ib PantHER trial enrolling pretreated HER2+ mBC (*n* = 12). The most common of all grade AEs reported in more than half of patients were hyperglycemia, fatigue, nausea and hypertension. Disease stabilization at 16 weeks was seen in six patients receiving the combination treatment (50%) and did not appear to correlate with the presence of *PIK3CA* mutation in archival tumor tissue. Interestingly, *PIK3CA* mutation in circulating tumor DNA (ctDNA) was detected in all patients, including those who tested negative for the mutation in the tissue (50%) [72]. This finding suggests that the frequency of *PIK3CA* mutation in HER2+ BC may be underestimated by archival tissue evaluation, proposing liquid biopsy as a useful tool to better capture temporal heterogeneity, thus potentially expanding the proportion of patients that could benefit from targeted therapy [Table 1]. To improve activity while limiting toxicities, current research focuses on isoform-specific PI3Ki.

In a phase I trial, the PI3Kα inhibitor, alpelisib, was administered in combination with T-DM1 in patients with HER2+ mBC progressed on prior trastuzumab and taxanes. Albeit patients were enrolled regardless of *PIK3CA* mutation, half of them had tumors with aberrations in the PI3K pathway. Interestingly, even though 59% of G ≥ 3 AEs occurred (mainly rash, fatigue and hyperglycemia), these were noted as manageable. In this heavily pretreated population (median of three prior lines for metastatic disease), the ORR was 43%, the CBR was 71% and the median PFS (mPFS) was 8.1 months. Even patients (*n* = 10) previously treated and progressed on prior T-DM1 had an ORR of 30%, a CBR of 61%, with a mPFS of 6.2 months [73]. As previously stated, T-DM1 seems to be active in *PIK3CA* mutant BC; however, there is evidence that activation of the PI3K pathway may lead to acquired resistance to T-DM1 [74], as for trastuzumab, and its inhibition could restore sensitivity to this drug.

Alpelisib has also been evaluated in combination with trastuzumab and an anti-HER3 antibody (LJM716) in a phase I study enrolling patients with refractory *PIK3CA*-mutant HER2+ mBC [75]. The rationale for concurrent targeting of HER2, HER3 and PI3K was the stimulation of HER3 expression with a concomitant MAPK pathway activation caused by PI3K inhibition, possibly impairing the activity of single-agents PI3Ki [76]. However, this study was stopped early due to toxicity, primarily gastrointestinal, highlighting the need to develop strategies to overcome overlapping toxicity in combinatorial approaches.

The ALPHABET trial is currently investigating the possibility of de-escalating therapy in HER2+, *PIK3CA*-mutant mBC pretreated with trastuzumab, comparing trastuzumab plus alpelisib versus trastuzumab plus chemotherapy +/- fulvestrant (if HR+). Moreover, the detection of *PI3KCA* mutations will be evaluated by ctDNA, assessing its clinical utility in comparison to tissue detection. This study provides a strategy to overcome resistance related to hyperactivation of PI3K signaling, trying to improve patients’ quality of life by avoiding chemotherapy related AEs.

Similarly, copanlisib and alpelisib combined with trastuzumab and pertuzumab are currently under investigation in HER2+, *PIK3CA*-mutant mBC, but as a maintenance therapy after initial chemotherapy (NCT04108858 and NCT04208178, respectively).

## 6. Biological and Clinical Evidence of *PIK3CA* Mutations in Triple-Negative Breast Cancer

TNBC is characterized by absence of both HR expression and HER2 overexpression/gene amplification, representing the most aggressive subtype with the poorest prognosis. Chemotherapy has long represented the only available treatment option for this tumor. However, in the past few years, significant advances have been made with the introduction of immunotherapy, ADCs and targeted agents [77,78]. In particular, according to the DESTINY-Breast04 trial results, the FDA recently approved trastuzumab deruxtecan (T-DXd) for the treatment of the new subtype “HER2-low mBC” (defined as HER2 IHC: 1+ or 2+/FISH not amplified), including both patients with HR+ and HR- tumors [79,80]. Mutations in *PIK3CA* gene are detected in approximately 9% of TNBC, including triple negative metastatic recurrence from primary HR+ breast cancer that may retain PIK3CA mutation [81].

TNBC can be further classified according to gene expression analysis into the six Lehman’s subtypes: basal-like 1 (BL-1), basal-like 2 (BL-2), mesenchymal (M), mesenchymal stem-like (MSL), immunomodulatory (IM) and luminal androgen receptor (LAR). Among these subtypes, LAR and MSL carry *PIK3CA* mutations with higher frequency [82]. LAR cells partly depend on AR (androgen receptor) signaling, as demonstrated by the inhibition of cell viability and tumor growth when AR is targeted by bicalutamide, an AR antagonist, and are less likely to benefit from chemotherapy regimens [83]. In addition, LAR TNBC cell lines that were *PIK3CA* H1047R mutant, were also sensitive to NVP-BEZ235, a PI3K/mTOR inhibitor [82]. Given these pre-clinical data and considering that LAR TNBC have an increased PI3K activity, the simultaneous targeting of AR and *PIK3CA* is currently under investigation [Table 1] [84]. A phase I clinical trial of alpelisib and enzalutamide in AR-positive and PTEN-positive BC, including a cohort of TNBC, is currently ongoing (NCT03207529). In addition, the LAR subgroup of TNBC showed a high sensitivity to CDK4/6 inhibitors (CDK4/6i) in vitro and in vivo, and CDK4/6i sensitize *PIK3CA* mutant cancers to PI3Ki [85]. The phase Ib PIPA trial has assessed the combination of the ß-isoform sparing PI3Ki taselisib with palbociclib in mBC, including a cohort of TNBC selected for activating PI3K mutations, reporting good tolerability and promising anti-tumor activity [86].

As *PIK3CA* mutations result in the constitutive activation of the PI3K/AKT/mTOR pathway, they inhibit apoptosis and favor cell-growth, causing resistance to neoadjuvant chemotherapy in TNBC, as observed by Hu et al. [87].

Some evidence shows better outcomes in *PIK3CA*-mutant TNBCs [88,89,90], independently of the specific subtype (BL or not-BL). However, available data are still scarce and conflicting, so the role of *PIK3CA* as a prognostic biomarker still needs to be elucidated. An analysis of patients with TNBC treated in the GeparSixto trial [52] demonstrated a worse response to neoadjuvant anthracycline-based chemotherapy in those TNBC carrying the *PIK3CA* H1047R mutation, with a lower rate of pCR [91].

To better understand if targeting *PIK3CA* could improve clinical outcomes in TNBC, PI3Ki, especially *PIK3CA*-selective ones, have been evaluated in different clinical trials, either as single agents or in combinatorial regimens.

In a phase II trial enrolling TNBC patients, buparlisib demonstrated prolonged clinical benefit in a small subset of patients, with no confirmed objective responses and a mPFS of less than two months [92]. Other trials have confirmed the limited efficacy of single-agent inhibitors of the PI3K pathway in TNBC, regardless of the presence of PIK3CA/AKT/PTEN alterations [93,94]. Based on these clinical data, it seems that patients with TNBC do not benefit from PI3Ki, even when no selection was done according to the *PIK3CA* mutational status. Thus, the development of buparlisib as single agent was discontinued and the interest moved towards possible PI3Ki-combined regimens.

Taxanes are effective in advanced TNBC; however, resistance often develops through different mechanisms, including PI3K pathway hyperactivation. In the phase II/III BELLE-4 trial, HER2- BC patients not pretreated with chemotherapy in the advanced setting were randomized to receive buparlisib or placebo with paclitaxel. Since the addition of buparlisib to paclitaxel did not improve mPFS in either the overall population (9.2 vs. 8 months; HR 1.18, CI 95% 0.82–1.68) or in the PI3K pathway-activated tumors, the trial was interrupted for futility at the end of phase II. Among the 416 enrolled patients, roughly 25% had TNBC; in these patients the prognosis was even worse in the experimental arm, with a mPFS of 5.5 months with buparlisib compared to 9.3 months in the placebo group (HR 1.68, 95% CI 0.91–3.79). In the experimental group, duration of buparlisib was lower, suggesting that its toxicity profile may have compromised the administration of backbone chemotherapy. Buparlisib treatment was associated with higher incidence of serious AEs (30.2% compared to 20.9% in the placebo group) and more than 40% of patients treated with the combination experienced diarrhea, alopecia, rash, nausea, and hyperglycemia [95]. The toxicity profile of buparlisib was consistent with previous reports, once again underlining the pharmacological limitations of pan-PI3Ki compared to isoform-selective PI3Ki which may achieve improved efficacy with fewer side effects.

In a phase Ib trial, the combination of alpelisib with paclitaxel demonstrated a challenging safety profile in patients with advanced solid tumors. This trial was interrupted because of high grade toxicity in particular hyperglycemia, G3 acute renal failure and G4 leukopenia [96]. Differently, alpelisib showed promising activity along with a manageable toxicity profile in combination with nab-paclitaxel in patients with HER2- pretreated mBC (30% TNBC). In the overall population, a higher CBR and mPFS were observed among patients with PI3K pathway-activation compared to those without alterations (CBR = 100% vs. 68%, OR = 1.47, *p* = 0.013; mPFS 11.9 vs. 7.5 months, HR = 0.44, 95% CI: 0.21–0.93, *p* = 0.027). In the TNBC subgroup, unselected for PI3K status, the ORR was 60% [97]. These data led to the design of the ongoing phase III EPIK-B2 randomized trial, evaluating this combination as first- or second-line treatment in patients with *PIK3CA*-mutant or PTEN-oss advanced TNBC (NCT04251533). Additionally, regarding the neoadjuvant setting, a phase II trial testing alpelisib plus nab-paclitaxel is currently enrolling patients with anthracycline-refractory early-stage TNBC with *PIK3CA* or *PTEN* alterations (NCT04216472). The results of these trials will define the role of alpelisib in TNBC.

Beyond chemotherapy, the mTNBC treatment landscape has evolved considerably with the introduction of immune checkpoint inhibitors (ICIs) in the first line for PD-L1-positive tumors and PARP inhibitors (PARPi), for germline BRCA1/2 (gBRCA) mutation carriers [77]. Preliminary data support the molecular rationale for testing combination of PI3Ki with these agents.

In preclinical models, buparlisib sensitized BRCA-proficient TNBC to olaparib leading to homologous recombination deficiency through downregulation of BRCA1/2 expression, supporting the synergistic activity of buparlisib and olaparib [98,99]. A phase I trial evaluating this combination was conducted in advanced BC (*n* = 24, 54% of which were TNBC) and ovarian cancer (*n* = 46) patients, regardless of gBRCA mutation. This trial demonstrated promising anti-cancer activity in both gBRCA-mutant and wild-type patients and feasibility for the combination, though transaminase elevation and neuropsychiatric AEs were common and responsible for dose-limiting toxicities [100]. Randomized trials will be needed to further define the efficacy of the combination compared to PARPi alone.

Since some emerging data suggest that PTEN loss is associated with resistance to immunotherapy [101], the combination of inhibitors of PI3K/AKT/mTOR signaling with ICIs should be evaluated. Preclinical models showed an improvement in cancer growth inhibition by combining an anti-PD-L1 antibody with a PI3Kß inhibitor [102].

## 7. Conclusions

*PIK3CA* mutations and, in general, alterations in the PI3K/AKT pathway, play a major role in the biology of BC, which is better established in the HR+/HER2- BC subtype than in TNBC and HER2+ BC for which research efforts are still needed. Further studies would clarify the prognostic role of *PIK3CA* mutations in these BC subtypes. Regardless the influence on prognosis, inhibition of PI3K could represent an appealing strategy to prevent or reverse treatment resistance. A better management of the toxicity profile is crucial for any further step towards the integration of PI3Ki in routine clinical practice.

Furthermore, in the current era of “personalized medicine”, to correctly select patients suitable for targeted therapy, it is necessary to determine appropriate methods for detecting *PIK3CA* mutations, considering that NGS may broaden the spectrum of detectable alterations, specifically evaluating their pathogenicity and their “druggability” [42,103,104,105,106].

Moreover, even though several trials with PI3Ki are ongoing both in HER2+ BC and TNBC, it is necessary to keep on investigating, integrating clinical data with pre-clinical ones, especially after the recent introduction of new agents as standard treatments, such as T-DXd and sacituzumab govitecan, as it is still an open question as to whether *PIK3CA* mutations may influence their efficacy.

## Figures and Tables

**Figure 1 jpm-12-01793-f001:**
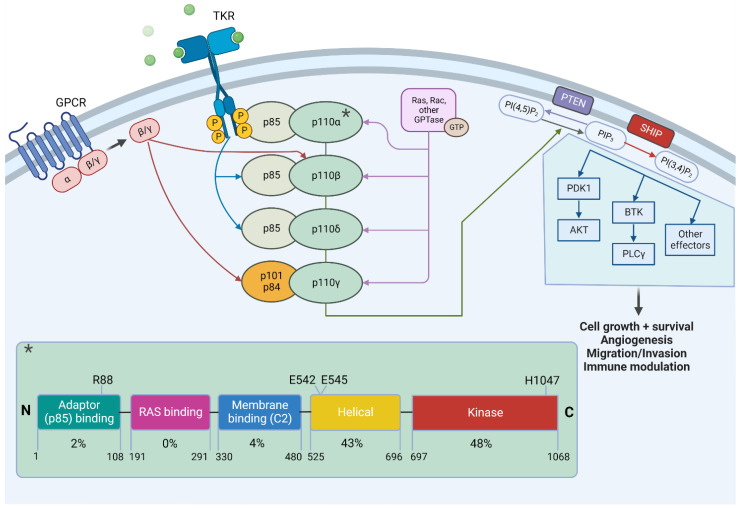
PI3K pathway, structure of the p110α kinase and most frequent mutations. Class I PI3Ks heterodimers consist of a catalytic subunit (four different isoforms, p110 α/β/δ/γ) and a regulatory one (p85 and p84 isoforms). The activation of TKRs or GPCR can trigger the PI3K-AKT-mTOR pathway. PI3K phosphorylates PIP2 to PIP3, activating the downstream effectors (including AKT) responsible for cell survival and growth. PTEN antagonizes the cascade by converting PIP3 to PIP2. In the bottom square, the five domains of p110α catalytic subunit* are detailed with their relative mutation occurrence rate (see text for details). PIP2, phosphatidyl inositol 4,5 bis-phosphate; PIP3, phosphatidyl inositol 3,4,5 bis-phosphate; GPCR, G protein-coupled receptors; TKR, tyrosine kinase receptors; Ras, rat sarcoma virus; Rac, ras-related C3 botulinum toxin substrate 1; GTP, guanosine-5′-triphosphate; AKT or protein kinase B (PKB); PTEN, phosphatase and tensin homolog; SHIP, Src homology 2 (SH2) domain containing inositol polyphosphate 5-phosphatase 1; PDK1, pyruvate dehydrogenase kinase 1; BTK, Bruton tyrosine kinase; PLC, phospholipase C. Figure created with BioRender.com. “*” is referred to the catalytic subtunit.

**Table 1 jpm-12-01793-t001:** Clinical trials of PI3K/AKT/mTOR inhibitors in HER2-positive and triple negative breast cancer.

PI3K Inhibitor	Combined Treatment	Mutation on PI3K/Akt/mTOR Pathway Related Genes Required	Phase	Setting	NCT Number/Study Name
Alpelisib ^α^	+Trastuzumab + pertuzumab	Yes, in part 2	III	Advanced HER2+ BC following induction with taxane	NCT04208178
					(EPIK-B2)
Copanlisib ^∞^	Trastuzumab + pertuzumab	Yes	I/II	Advanced HER2+ BC following induction with taxane	NCT04108858
Taselisib ^β^	Arm A: Taselisib with T-DM1	No	I	mBC, HER2+	NCT02390427
	Arm B: Taselisib with T-DM1 and pertuzumab				
	Arm C: Taselisib with pertuzumab and trastuzumab				
	Arm D: Taselisib with pertuzumab, trastuzumab and paclitaxel				
Copanlisib ^∞^	Trastuzumab	No	I/II	mBC, HER2+	NCT02705859
				≥1 line with Trastuzumab or T-DM1	(Panther)
BEZ235 (Dactolisib) ^¥^	+ Trastuzumab	No	I/II	mBC, HER2+ after Trastuzumab	NCT01471847
GDC-0084 ^×^	+ Trastuzumab	No	II	mBC, HER2+	NCT03765983
				with brain mets	
MEN1611 *	Trastuzumab +/− fulvestrant (if HR+)	Yes	I	mBC, HER2+,	NCT03767335 (B-PRECISE-01)
				>2 lines of anti-HER2	
MK2206 ^γ^	Trastuzumab and Lapatinib	No	I	Advanced HER2+ BC	NCT01705340
MK2206 ^γ^	Lapatinib	No	I	Advanced HER2+ BC	NCT01245205
Ipatasertib ^γ^	Trastuzumab + Pertuzumab + ET (if HR+)	Yes	I	Advanced HER2+ BC	NCT04253561
Inavolisib ^α^	Pertuzumab + Trastuzumab + Hyaluronidase + ET	Yes	II	Neoadjuvant HER2+, HR+ BC	NCT05306041
					(GeparPiPPa)
Alpelisib ^α^	Trastuzumab (+ET) vs chemotherapy + trastuzumab	Yes	III	mBC, HER2+/HR+	NCT05063786
				1 < line < 5	(ALPHABET)
Alpelisib ^α^	T-DM1	No	I	mBC, HER2+ after Trastuzumab-taxane regimen	NCT02038010
Alpelisib ^α^	+ trastuzumab + pertuzumab + chemotherapy	Yes	I	Neoadjuvant HER2+ BC	NCT04215003
XL147 (SAR245408, Pilaralisib) ^i^	Trastuzumab (+/- paclitaxel)	No	I/II	mBC, HER2+ after Trastuzumab	NCT01042925
Alpelisib ^α^	Tucatinib + fulvestrant	Yes	I/II	Advacnced BC, HER2+, >2 lines	NCT05230810
BKM120 (Buparlisib) ^i^	Lapatinib	Yes	I/II	Trastuzumab-resistant	NCT01589861 (PIKHER2)
				mBC, HER2+	
BKM120 (Buparlisib) ^i^	Trastuzumab + Paclitaxel	No	I II	Neoadjuvant HER2+	NCT01816594 (NeoPHOEBE)
Rapamycin	Inetetamab + chemotherapy vs. pyrotenib + chemotherapy	Yes	III	mBC, HER2+ after Trastuzumab	NCT04736589
Taselisibβ/Pictilisib ^i^	Palbociclib	Yes	I	mBC, HER2+ >2 lines	NCT02389842
				mTNBC >1 line	
AZD5363 (Capivasertib) ^γ^	/	Yes	I	mBC, ER+/HER2+	NCT01226316
Everolimus	Nab-paclitaxel	Yes	II	mTNBC	NCT04395989
Alpelisib ^α^	Nab-Paclitaxel	Yes or loss of function *PTEN*	II	Anthracycline refractory-TNBC, neoadjuvant	NCT04216472
Alpelisib ^α^	Sacituzumab govitecan	No	I	Advanced TNBC	NCT05143229
Alpelisib ^α^	Nab-paclitaxel	Yes or loss of function *PTEN*	III	Advanced TNBC, 2nd line	NCT04251533 (EPIK-B3)
Inavolisib ^α^	/	Yes	I	mTNBC	NCT03006172
CUDC907 (Fimepinostat) ^ç^	/	No	I	mTNBC	NCT02307240
Alpelisib ^α^	Enzalutamide	AR-positive and PTEN-positive	I	mTNBC	NCT03207529
Copanlisib ^∞^	Eribulin	No	I/II	Advanced TNBC,	NCT04345913
				1 < line ≤ 5	
PF-05212384 (Gedatolisib) ^#^	PTK-ADC	No	I	mTNBC,	NCT03243331
				≥2nd line	
Taselisib ^β^	/	Yes, without *KRAS* mutations or PTEN Loss	I	mTNBC	NCT04439175
INCB050465 (Parsaclisib) ^‡^	Pembrolizumab	No	I	mTNBC	NCT02646748
PQR309 (Bimiralisib) ^¥^	Eribulin	No	I	Advanced TNBC	NCT02723877 (PIQHASSO)
				1 < line ≤ 5	
Everolimus	Anti-PD1	No	I	mTNBC	NCT02890069
AZD5363 (Capivasertib) ^γ^	Paclitaxel	No	II	mTNBC	NCT02423603
AZD2014 (Vistusertib) ^<^	Selumetinib	Yes, or Ras/MEK pathway	II	mTNBC	NCT02583542
Taselisib ^β^	Enzalutamide	No	I/II	mTNBC, AR+	NCT02457910
Everolimus	AR-inhibitor	Yes and LAR subtype	I/II	mTNBC	NCT03805399
Alpelisib ^α^	/	Yes	II	mTNBC, ≥2nd line	NCT02506556 (PIKNIC)
Tenalisib ^¤^	/	No	II	mTNBC	NCT05021900
Ipatasertib ^γ^	Paclitaxel vs. placebo + paclitaxel	Yes	III	mTNBC, 1st line	NCT03337724
BKM120 (Buparlisib) ^i^ or Alpelisib ^α^	Olaparib	No	I	mTNBC, >1st line	NCT01623349
PF-05212384 (Gedatolisib) ^#^	PTK7-ADC	No	I	mTNBC, >1st line	NCT03243331
Ipatasertib ^γ^	+ paclitaxel +/-atezolizumab	No	III	mTNBC	NCT04177108
Eganelisib ^¶^	Atezolizumab + nab-paclitaxel	No	II	mTNBC, 1st line	NCT03961698 (MARIO-3)
Eganelisib ^¶^	Nivolumab	No	I	mTNBC	NCT02637531
AZD8186 ^†^	/	No	I	mTNBC	NCT01884285
AZD8186 ^†^	Docetaxel	Yes or loss of function *PTEN*	I	Advanced BC	NCT03218826
Ipatasertib ^γ^	/	Yes	II	mBC, ≤2 lines	NCT04591431
LOXO-783 ^§^	Alone or + paclitaxel	Yes, *PIK3CA* H1047R mutation	I	mBC, <5 lines	NCT05307705
AZD8835 ^∞^	/	Yes	I	mBC	NCT02260661
Taselisib ^β^/ Capivasertib ^γ^/ Copanlisib ^∞^	/	Yes, *PIK3CA* or *AKT*	I	mBC	NCT02465060
MK2206 ^γ^	Paclitaxel	No	I	mBC, ≤3 lines	NCT01263145
Alpelisib ^α^	/	Yes	I	mBC	NCT05238831
Inavolisib ^α^	/	Yes	II	mBC	NCT05332561
BEZ235 (Dactolisib) ^¥^	MEK162	No	I	mBC	NCT01337765
CYH33 ^α^	Olaparib	Yes or on DDR gene	I	mBC	NCT04586335
PF-05212384 (Gedatolisib) ^#^	Paclitaxel and carboplatin	No	I	mBC	NCT02069158
RLY-2608 ^α^	/	Yes	I	Advanced solid tumors	NCT05216432
Alpelisib ^α^	/	Yes	I	Solid tumors	NCT01219699

* PI3K α/β/γ inhibitor; ^γ^ AKT inhibitor; ^∞^ PI3K α/δ inhibitor; ^β^ PI3K α/δ/γ inhibitor; ^i^ PI3K α/β/δ/γ inhibitor; ^×^ PI3Kα and mTOR inhibitor, ^α^ PI3Kα inhibitor; ^¥^ PI3K α/β/δ/γ and mTOR inhibitor; ^†^ PI3K β/δ inhibitor; ^ç^ PI3K and HDAC (histone deacetylases) inhibitor; ^‡^ PI3Kδ inhibitor; ^¤^ PI3Kδ/γ inhibitor; ^¶^ PI3Kγ inhibitor; ^§^ PI3Kα H1047R inhibitor; ^#^ PI3Kα/γ and mTOR inhibitor; ^<^ pan mTOR inhibitor. Abbreviations: ET, endocrine therapy; HR, hormone receptor; ER, estrogen receptor; HER2, human epidermal growth factor receptor 2; TNBC, triple-negative breast cancer; BC, breast cancer; T-DM1, Trastuzumab emtansine; mBC, metastatic breast cancer; AR, androgen receptor; mTNBC, metastatic triple negative breast cancer; DDR, DNA damage repair.

## Data Availability

Not applicable.

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
