# Peer review of "PIK3CAMutations in Breast Cancer Subtypes Other Than HR-Positive/HER2-Negative"

_jpm, 2022, doi:10.3390/jpm12111793_

Round 1
Reviewer 1 Report
In this manuscript, Ascione et al. reviewed the literature regarding the PIK3CA mutations in breast cancer subtypes other than HR-positive/HER2-negative. Alpelisib, a selective p110α inhibitor, is approved for the treatment of hormone receptor (HR)+/HER2- PIK3CA mutant metastatic breast cancer (BC) that progressed to a first line endocrine therapy. PIK3CA mutation are also present in triple negative BC (TNBC) and HER2+ BC in which subtypes the role of PI3K inhibition is not well established. They summarized PI3K/AKT/mTOR pathway, describing most common mutations found in PI3K genes and how they can be detected. They describe the available biological and clinical evidence of PIK3CA mutations in breast cancers other than HR+/HER2-, summarizing clinical trials investigating PI3Ki in these subtypes. This review is important for HR-positive/HER2-negative breast cancer treatment. However, it is still required for explaining the following issues:
1. According to the latest research, a subset of patients with triple‑negative breast cancer is HER2-low expression breast cancer. Trastuzumab deruxtecan showed efficacy in patients with HER2-low metastatic breast cancer. Therefore, trastuzumab deruxtecan should be added in the review.
2. The authors should provide the detailed literature searching strategy in the supplementary information.
3. The adverse effects and predictive biomarkers for Alpelisib should be discussed.
4. Assessment of other potential predictive biomarkers for Alpelisib therapy in HR+ breast cancer would be very useful (PIK3CA mutation subtype, SNP, etc.).
5. The authors should add these references about the targeted therapy in breast cancer [PMID: 33685991, 35562334, 31699932, and 31969692] in the discussion section. Also, the discordance of PIK3CA mutations between paired primary and metastatic lesions should be discussed. Some references about this issue are recommended to cite [PMID: 29315431, 31969692].
6. There are still some grammatical and spelling errors throughout which could be easily corrected.
Author Response
Dear Reviewer,
We received your feedbacks. The authors would like to thank you for the time and efforts dedicated to our paper. The manuscript has been reviewed according to your suggestions that we believe would improve the quality of the work.
We also valued the efforts provided by the editorial office to enhance all the publication process.
We hope revised version will satisfy you, making the article suitable for publication.
We look forward to hearing from you.
Kind regards,
Carmen Criscitiello
Reviewer 1 Comments:
This review is important for HR-positive/HER2-negative breast cancer treatment.
Authors: we thank Reviewer 1 for finding our manuscript worthy of interest.
- According to the latest research, a subset of patients with triple‑negative breast cancer is HER2-low expression breast cancer. Trastuzumab deruxtecan showed efficacy in patients with HER2-low metastatic breast cancer. Therefore, trastuzumab deruxtecan should be added in the review.
Authors: we thank the reviewer for the suggestion and we totally agree that, given the great relevance of data deriving from the DESTINY-Breast04 trial, trastuzumab deruxtecan should be added. We have now included it (lines 431-435) and added the relative citation.
- The authors should provide the detailed literature searching strategy in the supplementary information.
Authors: at the moment of the submission of the revised version, we also added the literature searching strategy, as suggested.
- The adverse effects and predictive biomarkers for Alpelisib should be discussed.
Authors: we thank Referee 1 for raising this point. We discussed most frequent adverse effects related to alpelisib with the relative frequency, according to the data of the SOLAR-1 trial. We also reported how to treat and manage each of the described adverse effects (lines 251-262).
At the moment, only PIK3CA mutations represent a biomarker predictive of response to alpelisib even if, whether these mutations cause the activation of the PI3K/AKT/mTOR pathway seems to be irrelevant. Double PIK3CA mutations showed to increase the sensitivity to alpelisib (lines 222-226).
- Assessment of other potential predictive biomarkers for Alpelisib therapy in HR+ breast cancer would be very useful (PIK3CA mutation subtype, SNP, etc.).
Authors: apart from PIK3CA mutations, the role of PIK3CA copy number variants (CNVs) as a predictive biomarker still needs to be elucidated, even if, in small number of patients, they have not been associated with a better response to alpelisib. On the contrary, PTEN loss could confer resistance to alpelisib. In the revised version of the paper, we have now discussed these points, providing some evidence (lines 226-231).
- The authors should add these references about the targeted therapy in breast cancer [PMID: 33685991, 35562334, 31699932, and 31969692] in the discussion section. Also, the discordance of PIK3CA mutations between paired primary and metastatic lesions should be discussed. Some references about this issue are recommended to cite [PMID: 29315431, 31969692].
Authors: we thank the reviewer for the suggestion. We added the recommended references about target therapy in breast cancer, including an interesting article that highlights how the actionability of genomic alterations, according to different scales (e.g., ESCAT), should guide the choice of a target therapy, since only alterations with high actionability levels provide target treatments efficacy.
We also consider of value considering the spatial and temporal breast cancer heterogeneity, in fact we discussed it also underlining that the difference between primary and metastases exists not only in terms of presence/absence of PIK3CA mutations but also in terms of levels of mutation. In addition, PIK3CA mutations can be both gained or lost between the primary and the metastatic sites (lines 194-199).
- There are still some grammatical and spelling errors throughout which could be easily corrected.
Authors: we have reviewed the paper correcting grammatical and spelling errors.
Reviewer 2 Report
In this review article, Ascione L et al, have attempted to summarize the role of different mutations in PI3K genes in context of breast cancer. The authors have described the methods used for detecting these mutations along with relevant biological and clinical evidence supporting the importance of these mutations in breast cancer. They have provided information from different clinical trials that are evaluating efficacies and safety profile of drugs that could possibly target the PI3K/ AKT/ mTOR axis in breast cancer. The focus of the review is looking at the PI3K/ AKT pathway specifically in HER2+ and triple-negative breast cancer. Overall, the review article is well written and gives a good description of the latest developments in therapeutically targeting this pathway in breast cancer.
Author Response
Dear Editor,
We received the comments from Reviewer 2. The authors would like to thank the expert for the time and efforts dedicated to our paper.
Comments:
Overall, the review article is well written and gives a good description of the latest developments in therapeutically targeting this pathway in breast cancer.
Authors: we thank Reviewer 2 for its kind words on our manuscript.
We valued the efforts provided by the editorial office to enhance all the publication process.
We hope revised version will satisfy the Reviewers, making the article suitable for publication.
We look forward to hearing from you.
Kind regards,
Carmen Criscitiello
Round 2
Reviewer 1 Report
The revised manuscript has made a great improvement.